# NF-κB as an Inflammatory Biomarker in Thin Endometrium: Predictive Value for Live Birth in Recurrent Implantation Failure

**DOI:** 10.3390/diagnostics15141762

**Published:** 2025-07-12

**Authors:** Zercan Kalı, Pervin Karlı, Fatma Tanılır, Pınar Kırıcı, Serhat Ege

**Affiliations:** 1Private Gözde Hospital, 44100 Malatya, Turkey; 2Private Clinic, 55020 Samsun, Turkey; parpi2300@hotmail.com; 3Private Clinic, 21100 Diyarbakır, Turkey; fatmatanilir@hotmail.com; 4Gynecology and Obstetric Department, Turgut Özal University, 44210 Malatya, Turkey; pinarkiricidr@hotmail.com; 5Department of Obstetrics and Gynecology, Dicle University, 21070 Diyarbakır, Turkey; sehatege782@gmail.com

**Keywords:** NF-κB, thin endometrium, live birth

## Abstract

**Background:** Recurrent implantation failure (RIF) poses a major challenge in assisted reproductive technologies, with thin endometrium (≤7 mm) being a frequently observed yet poorly understood condition. Emerging evidence implicates nuclear factor-kappa B (NF-κB), a key transcription factor in inflammatory signaling, in impaired endometrial receptivity. However, its clinical relevance and prognostic value for live birth outcomes still need to be fully elucidated. **Objective:** We aim to evaluate the expression levels of endometrial NF-κB in patients with RIF and thin endometrium and to determine its potential as a predictive biomarker for live birth outcomes following IVF treatment. **Methods:** In this prospective case–control study, 158 women were categorized into three groups: Group 1 (RIF with thin endometrium, ≤7 mm, *n* = 52), Group 2 (RIF with normal endometrium, >7 mm, *n* = 38), and fertile controls (*n* = 68). NF-κB levels were assessed using ELISA and immunohistochemical histoscore. Pregnancy outcomes were compared across groups. ROC analysis and multivariable logistic regression were performed to assess the predictive value of NF-κB. **Results:** NF-κB expression was significantly elevated in Group 1 compared to Group 2 and controls (*p* = 0.0017). ROC analysis identified a cut-off value of 7.8 ng/mg for live birth prediction (AUC = 0.72, sensitivity 74%, specificity 75%). Multivariable analysis confirmed NF-κB is an independent predictor of live birth (*p* = 0.045). Histological findings revealed increased NF-κB staining in luminal and glandular epithelial cells in the thin endometrium group. **Conclusions:** Increased endometrial NF-κB expression is associated with thin endometrium and reduced live birth rates in RIF patients. NF-κB may serve not only as a biomarker of pathological inflammation but also as a prognostic tool for treatment stratification in IVF. Based on findings in the literature, the therapeutic targeting of NF-κB may represent a promising strategy to improve implantation outcomes.

## 1. Introduction

RIF is defined as the failure to achieve a clinical pregnancy after the transfer of good-quality embryos in at least two IVF or ICSI cycles. Despite advances in assisted reproductive technologies (ARTs), RIF remains a persistent challenge with multifactorial etiologies [1,2,3,4].

While early efforts focused primarily on ovarian and oocyte parameters, recent studies emphasize the role of the endometrium, particularly EMT, as a critical determinant of implantation success. Although there is no consensus on the cut-off value for thin EMT, measurements below 7–8 mm are generally associated with poorer outcomes [5,6,7]. Interestingly, live births have still been reported with EMT as low as 4 mm, suggesting that endometrial receptivity is influenced by factors beyond structural thickness alone [8,9,10]. Several factors, such as previous uterine surgeries, dilation and curettage, and the use of clomiphene citrate, have been associated with impaired endometrial thickening, though many cases remain unexplained [11,12]. While EMT is known to correlate with circulating estrogen levels, optimal proliferation may not occur even under adequate estrogen supplementation in some patients [12].

Disruptions in endometrial vascularization, excessive oxidative stress, and abnormal collagen accumulation have also been proposed as contributors to EMT thinning [13,14,15]. These structural and molecular alterations may compromise endometrial receptivity regardless of thickness.

In this context, the transcription factor nuclear factor-kappa B (NF-κB) has gained attention for its role in inflammatory and fibrotic processes within the endometrium. NF-κB plays a central role in regulating cytokine expression, immune signaling, and tissue remodeling during endometrial preparation [16,17,18,19,20,21]. Estrogen and progesterone normally modulate NF-κB activation in a controlled manner [22,23], but the overexpression of NF-κB has been implicated in chronic endometrial inflammation and periglandular fibrosis, particularly in endometriosis models [16,17].

Based on this background, we hypothesized that elevated NF-κB expression in the endometrium may be associated with thin EMT in women with RIF. Therefore, this study aimed to investigate the tissue expression levels and immunohistochemical staining characteristics of NF-κB in women diagnosed with RIF and an EMT < 7 mm.

## 2. Materials and Methods

### 2.1. Study Design and Participants

This prospective case–control study was conducted at the IVF unit of Gözde Hospital, in Malatya, Turkey, between January 2022 and August 2024. A total of 158 women were enrolled and stratified into three groups based on reproductive history and EMT measured on the day of human chorionic gonadotropin (hCG) administration:Group 1 (*n* = 52): Patients with RIF and thin endometrium (EMT ≤ 7 mm).Group 2 (*n* = 38): Patients with RIF and normal endometrium (EMT > 7 mm).Control Group (*n* = 68): Fertile women with proven fertility.

RIF was defined as the failure to achieve pregnancy following two or more unsuccessful IVF/ICSI attempts, in line with the current literature [1]. Controlled ovarian stimulation was performed using recombinant FSH (follitropin alfa or follitropin beta; Gonal-f^®^, (Merck Serono, Darmstadt, Germany) or Puregon^®^,( MSD, Kenilworth, NJ, USA) in combination with a GnRH antagonist protocol (cetrorelix acetate, Cetrotide^®^ (Merck Serono, Darmstadt, Germany) or Orgalutran^®^ (Organon, Oss, The Netherlands)To minimize the influence of embryo-related factors on implantation outcomes, all transfers were performed using morphologically good-quality embryos in accordance with standard clinical practice.

Endometrial thickness was assessed by transvaginal ultrasonography using a sagittal view of the uterus while the bladder was empty, with measurements obtained from the thickest echogenic area between the two basal layers, using a Voluson E8 system (GE Healthcare, Waukesha, WI, USA),ultrasound system. The measurement was taken from the thickest echogenic area extending from one stratum basalis to the other.

Patients undergoing hysteroscopy (H/S) due to RIF were evaluated in the secretory phase, and endometrial biopsies were obtained immediately afterward using a Pipelle cannula (CooperSurgical, Trumbull, CT, USA). Biopsy specimens were processed as follows:A.A portion was preserved in phosphate-buffered saline (PBS) and stored at –80 °C for subsequent NF-κB protein analysis using ELISA.B.The remaining tissue was fixed in 10% formalin for immunohistochemical analysis.

Patients with intrauterine pathologies such as endometrial polyps, submucosal fibroids, uterine septum, or adhesions diagnosed during hysteroscopy were excluded from the study. Additionally, women with a history of dilatation and curettage (D&C), endometrioma, hydrosalpinx, or clomiphene citrate use were also excluded to eliminate potential confounders. The flow of participant inclusion, exclusion, and group allocation is illustrated in Figure 1.

### 2.2. Clinical and Laboratory Assessments

The following clinical and laboratory parameters were recorded for all patients: age, body mass index (BMI), the duration of infertility, the number of previous failed IVF attempts, serum levels of estradiol, luteinizing hormone (LH), follicle-stimulating hormone (FSH), anti-Müllerian hormone (AMH), thyroid-stimulating hormone (TSH), antral follicle count, the duration of ovarian stimulation, the total number of metaphase II (MII) oocytes retrieved, the number of high-quality embryos, the number of transferred embryos, and endometrial thickness. Hormonal levels were measured using the chemiluminescent immunoassay method. Hormonal levels (FSH, LH, estradiol, progesterone) were measured using a chemiluminescent immunoassay method with the Cobas e411 analyzer (Roche Diagnostics, Mannheim, Germany).

EMT was measured by transvaginal ultrasonography on the day of hCG administration. Endometrial biopsies were obtained using a Pipelle cannula under hysteroscopic guidance. Hysteroscopies were performed during the mid-secretory phase of the menstrual cycle, timed to reflect the implantation window—either after embryo transfer in natural cycles or following hormonal preparation in artificial cycles to mimic luteal phase conditions. Immediately after the hysteroscopic examination, endometrial biopsies were obtained during the same session using a Pipelle cannula. This approach was chosen to evaluate endometrial NF-κB levels under conditions that closely resembled the physiological environment of implantation.

The control group consisted of 68 healthy women with regular menstrual cycles, a history of spontaneous conception, at least one live birth, and no history of infertility or assisted reproductive treatment. These participants were recruited on a voluntary basis during routine gynecological evaluations or contraceptive counseling visits. Endometrial biopsies were obtained during the mid-secretory phase of the natural menstrual cycle (approximately 7–10 days after ovulation, corresponding to the presumed window of implantation).

To exclude the possibility of chronic endometritis, plasma cell infiltration was evaluated in endometrial biopsy samples. For this purpose, immunohistochemical staining with CD138 was performed, and cases with positive results were excluded from the study. The presence of plasma cells was considered a histopathological diagnostic criterion for chronic endometritis.

### 2.3. Endometrial NF-κB Expression Analysis (Immunohistochemistry)

Paraffin-embedded sections (5 µm) were deparaffinized and rehydrated. Antigen retrieval was performed using microwave heating in citrate buffer. Endogenous peroxidase activity was blocked, and slides were incubated with NF-κB/p65 antibody. AEC was used for chromogenic detection, and Mayer’s hematoxylin was used for counterstaining. Positive and negative controls were included. NF-κB expression was scored using a histoscore: HS = Extent × Intensity.

Extent: 0.1 (<25%), 0.4 (26–50%), 0.6 (51–75%), 0.9 (76–100%). Intensity: 0 (none), 0.5 (very weak), 1 (low), 2 (moderate), 3 (strong).

### 2.4. Endometrial NF-κB Protein Measurement via ELISA

Endometrial samples were rinsed with PBS and weighed. After homogenization and freeze–thaw cycles, samples were centrifuged and supernatants collected. NF-κB levels were measured using a commercial ELISA kit (Sunred Bioscience, Sunred Bioscience, Shanghai, China; Cat. No. 201-12-3456) and normalized to tissue weight (ng/mg). Detection range: 0.15–40 ng/mL. Intra- and inter-assay CVs: <10% and <12%, respectively.

### 2.5. Outcome Measures

Biochemical pregnancy, clinical pregnancy, and live birth rates were compared across groups. Biochemical pregnancy was defined as a positive serum β-hCG test (≥5 mIU/mL) measured approximately 12–14 days after embryo transfer, without the subsequent ultrasound evidence of a gestational sac.

Clinical pregnancy was defined as the presence of a gestational sac on transvaginal ultrasound, with or without fetal cardiac activity. For analytical purposes, clinical pregnancy included both cases that progressed to live birth and those that resulted in miscarriage (abortus).

Live birth was defined as the delivery of at least one viable infant after 24 weeks of gestation.

These outcomes were verified through the e-Nabız national health registry system, using patient-specific hospital or governmental records and ensuring the accurate confirmation of live birth status.

### 2.6. Statistical Analysis

Statistical analyses were performed using IBM SPSS Statistics version 25.0 (IBM Corp., Armonk, NY, USA). Normality was assessed by the Kolmogorov–Smirnov test. ANOVA or Kruskal–Wallis tests were used for group comparisons. Categorical variables were analyzed with Chi-square or Fisher’s exact test. ROC analysis determined NF-κB cut-off (7.8 ng/mg) for live birth (AUC = 0.72). Significance was set at *p* < 0.05.

## 3. Results

The demographic and clinical characteristics of the study groups are presented in Table 1. No statistically significant differences were observed between the groups in terms of age, BMI, or hormone levels (FSH, LH, estradiol, TSH and AMH) (*p* > 0.05). These findings indicate that the groups were comparable with respect to age and baseline hormonal profiles.

The duration of infertility and the number of previous failed IVF attempts were reported for only Group 1 and Group 2, and no significant differences were found between these two groups.

Additionally, antral follicle count, stimulation duration, the total number of MII oocytes retrieved, rge number of high-quality embryos, and the number of embryos transferred did not differ significantly among the groups. These results suggest a homogenous ovarian response and embryo quality across the study population.

Endometrial thickness differed significantly among the groups (*p* < 0.0001). The mean endometrial thickness was markedly lower in Group 1 (4.32 mm), whereas it was within the physiological range in Group 2 (9.21 mm) and the control group (9.45 mm). This finding may indicate a potential structural limitation in Group 1 that could be associated with reduced implantation potential.

The evaluation of pregnancy outcomes among the three groups revealed statistically significant differences in some parameters.

Biochemical pregnancy rates were 11.5% in Group 1, 5.3% in Group 2, and 4.4% in the control group. However, the differences among the groups were not statistically significant (*p* = 0.1454), suggesting that there was no major discrepancy in the early implantation stage.

In contrast, clinical pregnancy rates showed a statistically significant difference among the groups (*p* = 0.0314). Group 1 had a substantially lower rate (9.6%) compared to Group 2 (31.6%) and the control group (35.3%), indicating reduced endometrial support for embryo implantation in Group 1.

Abortus rates were similar across all groups and did not differ significantly (*p* = 0.9849), suggesting that the risk of pregnancy loss after clinical implantation was comparable among the groups.

Live birth rates were significantly lower in Group 1 (3.8%) compared to Group 2 (23.7%) and the control group (27.9%) (*p* = 0.0141). This highlights the markedly impaired ability in Group 1 to maintain and complete a viable pregnancy.

Similarly, the proportion of non-pregnant patients was highest in Group 1 (78.8%) and significantly greater than in Group 2 (63.2%) and the control group (60.3%) (*p* = 0.0035), reflecting the overall lower IVF success rate in Group 1.

Taken together, these findings suggest that Group 1 had significantly poorer IVF outcomes, which may be associated with reduced endometrial thickness and elevated NF-κB levels observed in this group (Table 2).

The diagnostic performance of endometrial NF-κB levels in predicting live birth outcomes was evaluated using receiver operating characteristic (ROC) curve analysis. The dependent variable was defined as a binary outcome, indicating the presence or absence of a live birth. The ROC curve was constructed by plotting sensitivity and specificity values across a range of threshold levels.

The area under the ROC curve (AUC) was calculated as 0.72, indicating the moderate discriminative ability of endometrial NF-κB levels for predicting live birth. This AUC value suggests that the predictive model performs significantly better than chance.

The optimal cut-off value for NF-κB was determined to be 7.8 ng/mg based on the Youden Index (sensitivity + specificity − 1). At this threshold, the sensitivity was 74%, and the specificity was 75%. The red cross on the ROC curve indicates the optimal cut-off point where diagnostic accuracy was maximized (Figure 2).

Table 3 compares the clinical and biochemical characteristics of patients who achieved live birth (*n* = 30) and those whose pregnancies did not result in live birth (*n* = 22), selected from the entire cohort of patients included in the IVF and control groups. The second group consists of individuals with positive serum β-hCG levels and evidence of implantation, but whose pregnancies either ended in miscarriage or remained limited to biochemical pregnancy.

One of the most striking differences between the groups was observed in NF-κB levels. The mean NF-κB level was significantly lower in the live birth group (4.90 ± 2.34 ng/mg) compared to the group without live birth (11.28 ± 5.60 ng/mg; *p* = 0.0004). This finding suggests that elevated NF-κB levels may reflect a proinflammatory endometrial environment, potentially impairing the progression of pregnancy.

Similarly, endometrial thickness differed significantly between the groups. The mean endometrial thickness was 9.21 ± 3.42 mm in the live birth group and 4.32 ± 1.28 mm in the non-live birth group (*p* < 0.0001). This underscores the importance of sufficient endometrial thickness as a key determinant for the maintenance and successful continuation of pregnancy.

In contrast, no statistically significant differences were observed between the groups in terms of age, BMI, FSH, LH, estradiol, AMH, or TSH levels (*p* > 0.05). These results indicate that, among patients who achieved pregnancy, local endometrial factors—particularly tissue structure and inflammation—may play a more critical role in determining live birth outcomes than demographic or systemic hormonal parameters.

As shown in Table 4, this multivariable logistic regression model was conducted to identify independent predictors of live birth among patients who achieved pregnancy. The model included clinical and biochemical variables such as age, BMI, AMH, FSH, LH, E2, NF-κB levels, and endometrial thickness.

Among all variables, the NF-κB level emerged as a statistically significant independent predictor of live birth (B = −0.204, *p* = 0.045). The negative coefficient indicates that higher NF-κB expression is associated with a reduced likelihood of live birth. Specifically, each unit increase in NF-κB level was associated with an 18.5% decrease in the odds of live birth (OR = 0.815; 95% CI: 0.66–1.00).

Although endometrial thickness showed a positive association with live birth (OR = 1.178), it did not reach statistical significance (*p* = 0.222). Similarly, AMH and FSH levels also demonstrated a trend toward a positive relationship with live birth (ORs of 1.527 and 1.380, respectively), but these associations were not statistically significant (*p* = 0.272 and *p* = 0.130).

BMI and LH levels approached the threshold for statistical significance (*p* = 0.099 and *p* = 0.055, respectively), suggesting that these factors may have a potential influence on live birth outcomes. Notably, LH showed a trend toward a negative association with live birth (OR = 0.748).

Other variables, including age and E2, did not demonstrate any significant associations with live birth (*p* > 0.25), indicating the limited predictive value of this model.

As shown in Figure 3**,** immunohistochemical analysis revealed the marked difference in NF-κB expression between endometrial tissues of differing thickness. In the EMT ≤ 7 mm group (Figure 3A), strong nuclear and cytoplasmic NF-κB staining was observed in glandular epithelial cells, as indicated by red arrows. In contrast, tissues with EMT > 7 mm (Figure 3B) showed significantly weaker staining intensity, suggesting reduced NF-κB activity in thicker endometrial samples. These histological findings support the quantitative results, indicating that elevated NF-κB expression is associated with thinner endometrium, which may reflect an inflammatory endometrial microenvironment unfavorable for implantation.

## 4. Discussion

Embryo quality, although essential, is not solely sufficient to ensure successful implantation. The structural and functional integrity of the endometrium plays a pivotal role in facilitating embryo attachment and invasion, and ultimately achieving a viable pregnancy. In this context, thin endometrium (≤7 mm), particularly in women with recurrent implantation failure (RIF), represents a clinically significant issue, as it may reflect suboptimal endometrial receptivity. The reported prevalence rates of thin endometrium among RIF patients vary widely from 1% to 66% [24], and several studies have consistently demonstrated the condition’s association with lower implantation and live birth rates. Although the pathophysiological mechanisms are not fully elucidated, impaired vascularization and increased stromal fibrosis have been suggested as potential contributing factors [24].

An increasing body of evidence indicates that inflammation plays a critical role in regulating endometrial receptivity, with NF-κB emerging as a central mediator. NF-κB acts as a transcriptional regulator involved in cytokine production, immune cell recruitment, and tissue remodeling—processes that are all vital during the peri-implantation period. Several studies have demonstrated that NF-κB activation in endometrial epithelial cells contributes to early embryo–endometrium communication and modulates gene expression profiles necessary for implantation [25,26,27]. While basal NF-κB activity supports a pro-receptive inflammatory milieu, its overactivation may lead to pathological inflammation, resulting in disrupted endometrial function [28]. This dual role could explain the divergent findings reported in the literature. For instance, Erşahin [29] and Wu [30] suggested that excessive NF-κB expression impairs endometrial receptivity, whereas van Mourik [31] and Robertson [32] highlighted its beneficial role in promoting immune tolerance and supporting trophoblast invasion.

Maintaining NF-κB activity within a physiological range appears to be essential for the fine balance between controlled inflammation and tissue stability. Persistent NF-κB signaling has been implicated in several pathological processes observed in endometriosis, including chronic inflammation, aberrant cell proliferation, enhanced angiogenesis, and abnormal cellular invasiveness. Furthermore, NF-κB has been associated with oxidative stress, epithelial-to-mesenchymal transition, and fibrotic transformation, underscoring its multifaceted role in uterine pathophysiology [33]. The clinical relevance of this mechanism becomes particularly evident in endometriosis-related cases; recent evidence indicates that NF-κB plays a role in the pathogenesis of the disease by disrupting endometrial structure through an inflammatory microenvironment, thereby impairing implantation [34,35].

Our findings further support this hypothesis, showing significantly increased NF-κB expression levels in both the RIF patients with a thin endometrium and RIF with normal endometrium groups compared to healthy controls. Specifically, NF-κB concentrations were 11.28 ± 5.60 ng/mg and 4.90 ± 2.34 ng/mg in Group 1 and Group 2, respectively, while the control group demonstrated levels of 3.76 ± 0.41 ng/mg (*p* = 0.0017). Histoscore analysis also confirmed elevated NF-κB expression among patients with RIF (1.20 ± 0.54 in Group 1 and 0.55 ± 0.81 in Group 2) compared to controls (0.95 ± 0.45; *p* = 0.0478), indicating a heightened proinflammatory state and disturbed endometrial immune environment [36,37].

Importantly, NF-κB levels were found to correlate with clinical pregnancy outcomes, specifically live birth rates, which represent the most clinically relevant endpoint in ART [38,39]. Through ROC analysis, a diagnostic cut-off value of 7.8 ng/mL was identified, yielding an AUC of 0.72 with a sensitivity of 74% and a specificity of 75%. These findings suggest that NF-κB may serve not only as a molecular indicator of pathological inflammation but also as a potential biomarker for predicting live birth outcomes. While previous studies have primarily focused on intermediate endpoints such as biochemical pregnancy or implantation, this study provides preliminary evidence of an association between endometrial immune activity and the ultimate clinical goal of live birth. However, further comprehensive, mechanistic, and prospective studies are needed to validate this relationship and determine its applicability in clinical practice.

Although the role of NF-κB in the implantation process has been previously described in the literature [40], studies that quantitatively assess its diagnostic value and define specific cut-off thresholds remain limited. In this study, the demonstrated association between NF-κB levels and live birth offers a novel perspective on the immunological dynamics of endometrial receptivity, particularly through analytical tools such as ROC curve analysis. In this context, NF-κB may be considered not only a mechanistic biomarker but also a potential therapeutic target in future research aimed at optimizing ART.

Notably, although traditional markers such as age, BMI, and ovarian reserve indicators (FSH, AMH, AFC) were comparable between groups (*p* > 0.1 for all), endometrial thickness and NF-κB levels emerged as the only parameters showing statistically significant associations with live birth, providing preliminary data that may serve as a basis for future research. This reinforces the notion that embryo-independent uterine factors—especially local immune modulation—may be more predictive of IVF success in RIF populations.

NF-κB appears to be open to further investigation, not only as a potential biomarker but also as a modifiable mechanism in future studies. Given the strong inverse relationship between NF-κB activity and live birth, strategies aiming to modulate endometrial inflammation—such as intrauterine anti-inflammatory agents or systemic modulators—may offer therapeutic benefit, particularly in patients with persistently thin endometrium and elevated NF-κB expression.

According to existing evidence in the literature, the immunometabolic effects of nutraceuticals such as myo-inositol may modulate endometrial inflammation and, consequently, improve treatment outcomes [41]. In this context, exploring additional strategies to target molecular pathways such as NF-κB is of great importance for developing personalized therapeutic approaches to enhance implantation success.

## 5. Conclusions

The observed association between NF-κB levels and live birth rates suggests that this molecule may hold clinical relevance in the endometrial context. This finding supports the existence of possible interaction between the immunological state of the endometrium and pregnancy success. However, our study is limited to observational data, and further mechanistic and interventional research is needed to draw causal conclusions. In this regard, our findings provide a hypothesis-generating basis for future investigations.

## Figures and Tables

**Figure 1 diagnostics-15-01762-f001:**
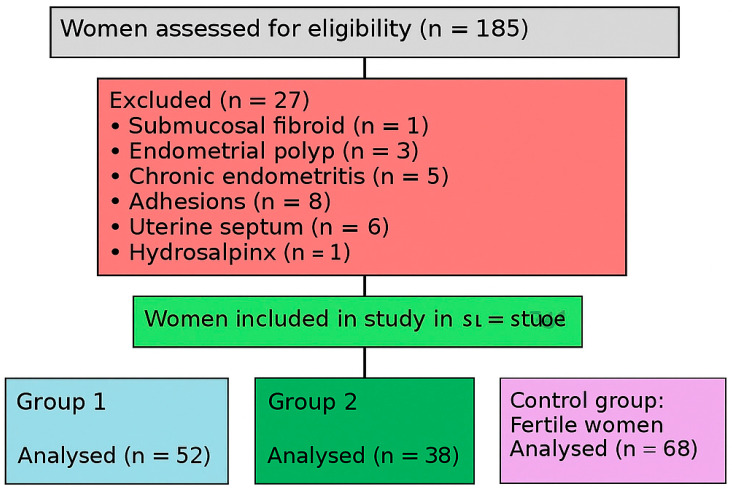
Flowchart of participant inclusion and group allocation.

**Figure 2 diagnostics-15-01762-f002:**
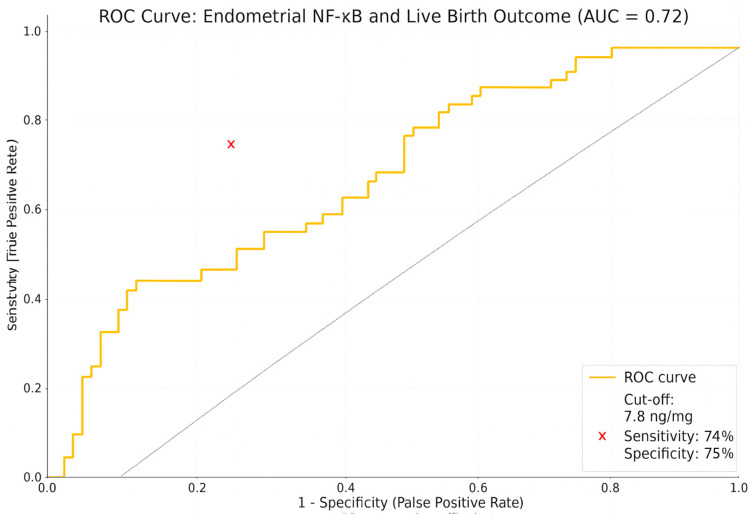
ROC curve analysis of endometrial NF-κB levels for predicting live birth in IVF-treated RIF patients. Receiver operating characteristic (ROC) curve illustrating diagnostic performance of endometrial NF-κB levels in predicting live birth outcomes among women with recurrent implantation failure (RIF). AUC was 0.72, indicating moderate predictive accuracy. Optimal cut-off value for NF-κB was determined as 7.8 ng/mg, yielding sensitivity of 74% and specificity of 75%. Red cross marks optimal cut-off point on curve.

**Figure 3 diagnostics-15-01762-f003:**
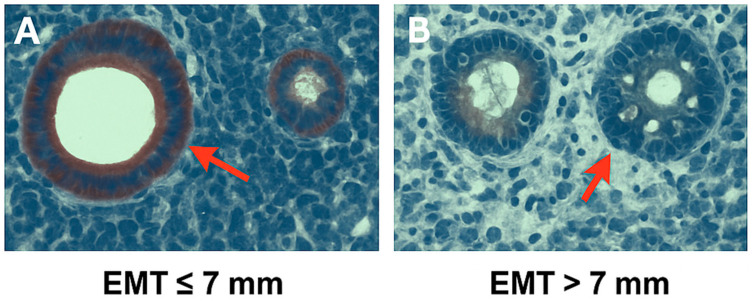
Histological comparison of endometrial thickness in groups. Representative histological sections of endometrial tissue in patients with different EMT values. (**A**) EMT ≤ 7 mm, showing compact glandular structure. (**B**) EMT > 7 mm, demonstrating more proliferative morphology. Red arrows indicate endometrial glands. Magnification: ×400. Scale bar = 50 μm.

**Table 1 diagnostics-15-01762-t001:** Demographic and laboratory characteristics of study participants across three groups.

Variable	Group 1 (*n* = 52)	Group 2 (*n* = 38)	Control (*n* = 68)	*p*-Value
Age (years)	28.76 ± 6.70	29.21 ± 8.32	30.12 ± 7.40	0.34
BMI (kg/m^2^)	22.33 ± 5.11	22.41 ± 5.40	22.62 ± 5.20	0.11
Duration of infertility (years)	3.43 ± 2.30	3.76 ± 2.87	–	–
Failed IVF attempts (*n*)	3.22 ± 1.44	3.02 ± 1.09	–	–
Estradiol (pg/mL)	35.6 ± 6.12	34.9 ± 5.41	45.7 ± 4.80	0.66
LH (mIU/mL)	4.95 ± 2.30	5.11 ± 3.22	5.47 ± 2.90	0.41
FSH (mIU/mL)	5.13 ± 3.01	5.22 ± 3.05	5.36 ± 2.85	0.39
AMH (ng/mL)	2.23 ± 1.47	1.86 ± 1.31	2.47 ± 0.94	0.1164
TSH (uIU/mL)	2.47 ± 0.64	2.70 ± 0.70	1.00 ± 0.31	0.1266
Antral Follicle Count	9.32 ± 5.09	11.64 ± 7.67	–	–
Stimulation Duration (days)	10.16 ± 1.83	11.82 ± 3.19	–	–
Total MII Oocyte Count	6.00 ± 4.03	7.96 ± 5.91	–	–
High-Quality Embryo Count	3.42 ± 2.17	3.79 ± 2.51	–	–
Transferred Embryo Count	1.11 ± 0.32	1.04 ± 0.19	–	–
EMT (mm)	4.32 ± 1.28	9.21 ± 3.42	9.45 ± 2.06	0.0000 *
NF-κB Level	11.28 ± 5.60	4.90 ± 2.34	3.76 ± 0.41	0.0017 *
NF-κB Histoscore	1.20 ± 0.54	0.55 ± 0.81	0.95 ± 0.45	0.0478 *

Note: Data are presented as mean ± standard deviation. Asterisk (*) indicates statistical significance at *p* < 0.05. “–” indicates that the data are not available or not applicable for the respective group.

**Table 2 diagnostics-15-01762-t002:** Distribution of IVF outcomes and their percentages across study groups.

Outcome Type	Group 1 (*n* = 52)	Group 2 (*n* = 38)	Control (*n* = 68)	*p*-Value
Biochemical pregnancy	6 (11.5%)	2 (5.3%)	3 (4.4%)	0.1454
Clinical pregnancy	5 (9.6%)	12 (31.6%)	24 (35.3%)	0.0314 *
Abortus	3 (5.8%)	3 (7.9%)	5 (7.4%)	0.9849
Live birth	2 (3.8%)	9 (23.7%)	19 (27.9%)	0.0141 *
Non pregnant	41 (78.8%)	24 (63.2%)	41 (60.3%)	0.0035 *

Note: Data are presented as number of cases with corresponding percentages in parentheses. *p*-values indicate comparisons among three groups, calculated using chi-square test. Asterisk (*) denotes statistically significant *p*-values (*p* < 0.05). Clinical pregnancy was defined as sum of live birth and abortion cases.

**Table 3 diagnostics-15-01762-t003:** Comparison of clinical, hormonal, and endometrial parameters between patients with and without live birth outcomes.

Variable	With Live Birth (*n* = 30)	Without Live Birth (*n* = 22)	*p*-Value
NF-κB (ng/mg)	4.90 ± 2.34	11.28 ± 5.60	0.0004 *
Age (years)	29.21 ± 8.32	28.76 ± 6.70	0.340
BMI (kg/m^2^)	22.41 ± 5.40	22.33 ± 5.11	0.110
FSH (mIU/mL)	5.22 ± 3.05	5.13 ± 3.01	0.390
LH (mIU/mL)	5.11 ± 3.22	4.95 ± 2.30	0.410
E2 (pg/mL)	34.9 ± 5.41	35.6 ± 6.12	0.660
AMH (ng/mL)	1.86 ± 1.31	2.23 ± 1.47	0.1164
TSH (uIU/mL)	2.70 ± 0.70	2.47 ± 0.64	0.1266
EMT (mm)	9.21 ± 3.42	4.32 ± 1.28	0.0000 *

Note: Data are presented as mean ± standard deviation. *p*-values were calculated using independent samples t-test or Mann–Whitney U test based on data distribution. Asterisk (*) indicates statistical significance at *p* < 0.05. “Without Live Birth” group includes patients with confirmed pregnancy (positive serum β-hCG) who experienced biochemical pregnancy loss or clinical miscarriage.

**Table 4 diagnostics-15-01762-t004:** Multivariable logistic regression analysis of factors associated with live birth.

Variable	Coefficient (B)	Standard Error	*p*-Value	Odds Ratio (exp(B))	95% CI for B	95% CI for OR
NF-κB level	−0.204	0.106	0.045 *	0.815	−0.413–0.005	0.66–1.00
EMT	0.164	0.134	0.222	1.178	−0.099–0.427	0.91–1.53
AMH (ng/mL)	0.423	0.386	0.272	1.527	−0.333–1.179	0.72–3.25
Age	−0.031	0.103	0.766	0.970	−0.233–0.172	0.79–1.19
BMI	−0.204	0.123	0.099	0.816	−0.445–0.038	0.64–1.04
FSH (mIU/mL)	0.322	0.213	0.130	1.380	−0.095–0.739	0.91–2.09
LH (mIU/mL)	−0.290	0.151	0.055	0.748	−0.587–0.007	0.56–1.01
E2 (pg/mL)	−0.001	0.013	0.917	0.999	−0.027–0.024	0.97–1.02

Note: Multivariable logistic regression analysis was performed to identify independent predictors of live birth. Odds ratios (OR) with 95% confidence intervals (CI) are presented for each variable. Asterisk (*) indicates statistical significance at *p* < 0.05.

## Data Availability

The datasets generated and/or analyzed during the current study are available from the corresponding author on reasonable request.

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
