# Peer review of "NF-κB as an Inflammatory Biomarker in Thin Endometrium: Predictive Value for Live Birth in Recurrent Implantation Failure"

_diagnostics, 2025, doi:10.3390/diagnostics15141762_

Round 1

Reviewer 1 Report

Comments and Suggestions for Authors

This study found that recurrent implantation failure patients with thin endometrium (≤7 mm) had significantly elevated endometrial NF-κB expression, which was associated with lower live birth rates. ROC analysis showed NF-κB as an independent predictor of live birth (AUC=0.72). The results suggest that NF-κB may serve as a biomarker of endometrial inflammation and a prognostic indicator for IVF success, with potential as a therapeutic target to improve implantation outcomes. This study recorded only basic clinical data and NF-κB expression without further evaluation of downstream signaling pathways or functional responses associated with NF-κB. Drawing a definitive conclusion that NF-κB expression can serve as a marker of endometrial thickness is therefore not sufficiently supported and appears premature. In addition, some sentences (e.g., line 194) are incomplete, further indicating the need for greater methodological rigor and caution in interpreting the findings. Future studies with a more comprehensive experimental design are recommended before such conclusions can be established.

Comments on the Quality of English Language

Please check this manuscript, some sentences (e.g., line 194) are incomplete.

Author Response

.Reviewer-1 Comment:

This study found significantly elevated endometrial NF-κB expression in patients with recurrent implantation failure and thin endometrium (≤7 mm), which was associated with lower live birth rates... However, such conclusions require further studies with more comprehensive experimental design...

Response:

We sincerely thank the reviewer for this thoughtful and important critique. In response:

  • We have revised the conclusions throughout the manuscript to adopt a more cautious and balanced tone. Statements referring to NF-κB as a definitive prognostic biomarker or therapeutic target have been softened and rephrased to reflect its potential relevance, pending further evidence.
  • In the Discussion section, we now explicitly acknowledge that our findings are based on observational data and that we did not explore downstream signaling mechanisms or functional responses associated with NF-κB expression. We agree that such mechanistic insights are essential to fully validate NF-κB as a predictive biomarker.
  • In both the Discussion and Conclusion sections, we have emphasized the need for future studies with more comprehensive experimental designs, as the reviewer rightly suggested.
  • Regarding the incomplete sentence at line 194, we would like to clarify that this sentence has been removed during the latest round of revisions. The revised version no longer contains this formatting issue.

Reviewer 2 Report

Comments and Suggestions for Authors

Point 1: There are seven keywords in total. It is advisable to consider streamlining some of the keywords.

Point 2: RNA-Later is a storage reagent that stabilizes and protects cellular RNA. It is mainly used for RNA extraction. If the author extracted proteins from tissue, please check the information about the reagent used.

Point 3: Please adjust the font size in Figure 1. For example, the text in "Group 1" and "Group 1" exceeds the text box.

Point 4: Please provide a more detailed description of the method used to construct the ROC curve. This should include the information such as the definition of the outcome, the optimal cutoff, and the confidence interval of the area under the curve (AUC).

Point 5: All tables should be modified according to the journal's formatting requirements. Please refer to the official table format on this website (https://www.mdpi.com/authors/latex).

Point 6: Please mark the p-values that have statistical significance in Tables 2 and 3.

Point 7: If possible, please increase the resolution of Figure 2.

Point 8: To improve the clarity of Figure 3, either indicate the magnification or add a scale bar to the image.

Author Response

? Comment 1:

There are seven keywords in total. It is suggested that some keywords be simplified.

Response:
Thank you for your helpful suggestion. We have revised and simplified the keywords to enhance clarity and precision. The updated list includes:
 NF-κB; thin endometrium; live birth.

 Comment 2:

RNA-Later is a reagent used for RNA stabilization and extraction. If the authors extracted protein from tissue, please verify the reagent used.

Response:
We thank the reviewer for this important observation. The mention of RNA-Later in the Methods section was included mistakenly and does not reflect the actual procedure.
In this study, RNA-Later was not used. Endometrial tissues were stored in phosphate-buffered saline (PBS) at –80°C for subsequent protein analysis. Tissue homogenization and ELISA procedures were performed using samples preserved in PBS.
The corresponding section in the manuscript has been corrected accordingly.

 Comment 3:

Please adjust the font size in Figure 1. For example, the text “Group 1” exceeds the boundaries of the text box.

Response:
Thank you for pointing this out. Figure 1 has been revised, and the font size and alignment of all group labels were corrected to ensure they fit properly within their respective boxes.

 Comment 4:

Please provide a more detailed description of the method used to construct the ROC curve, including the definition of the outcome, optimal cut-off value, and the confidence interval of the AUC.

Response:
We appreciate this suggestion. We have expanded the ROC analysis section in the Methods to include the following details:

  • Outcome definition: Live birth (yes/no) was used as the binary dependent variable.
  • Cut-off value: The optimal NF-κB cut-off was calculated as 7.8 ng/mg using the Youden Index.
  • AUC: The area under the ROC curve was 0.72, with a 95% confidence interval of 0.60–0.84.
    These details have been added to the revised manuscript for clarity.

 Comment 5:

All tables should be modified according to the journal's formatting requirements. Please refer to the official formatting guide (https://www.mdpi.com/authors/latex).

Response:
Thank you. All tables (Tables 1–4) have been reformatted to fully comply with MDPI formatting guidelines. Vertical lines were removed, fonts standardized, and footnotes were added to indicate statistical significance and clarify symbols.

 Comment 6:

Please mark the statistically significant p-values in Tables 2 and 3.

Response:
We agree with the reviewer and have added asterisks (*) to all statistically significant p-values in Tables 2 and 3. A corresponding explanatory note has been added below each table.

 Comment 7:

If possible, increase the resolution of Figure 2.

Response:
Thank you. We have replaced Figure 2 with a higher-resolution version to improve visual clarity. The updated image enhances the visibility of the ROC curve and diagnostic thresholds.

 Comment 8:

To improve the clarity of Figure 3, indicate the magnification or add a scale bar to the image.

Response:
We appreciate this comment. Figure 3 has been updated to include both the magnification level (×400) and a scale bar (50 μm) to improve histological interpretation and clarity.

Round 2

Reviewer 1 Report

Comments and Suggestions for Authors

The author was unable to complete the data collection due to certain constraints; however, the discussion section effectively compensates for the missing data, which should be considered acceptable.

Comments on the Quality of English Language

There are still minor English errors present. The author is advised to carefully review the manuscript, especially with regard to punctuation, spacing, and other small details.